# LINC01117 inhibits invasion and migration of lung adenocarcinoma through influencing EMT process

Linjun Liu[1☯], Wenjia Ren[1☯], Licheng Du[1], Ke Xu[2]*, Yubai Zhou[1]*

**1** Department of Environment and Life Sciences, Beijing University of Technology, Beijing, China, **2** NHC Key Laboratory of Biosafety, National Institute for Viral Disease Control and Prevention, Beijing, China

☯ These authors contributed equally to this work.
* xuke@ivdc.chinacdc.cn (KX); zhouyubai@bjut.edu.cn (YZ)

**Data Availability Statement:** All relevant data are within the manuscript and its Supporting Information files.

**Funding:** The author(s) received no specific funding for this work.

## Abstract

### Background

Studying the mechanism of action of LncRNAs in lung adenocarcinoma (LUAD) is of great importance for an in-depth understanding of the molecular mechanism of lung adeno carcinogenesis and development.

### Objective

The aim is to identify a long non-coding RNA LINC01117 that is specifically and highly expressed in LUAD cells and to investigate its biological functions and molecular mechanisms in LUAD cells, providing a new potential target for targeting LUAD therapy.

### Methods

This study used publicly available data downloaded from The Cancer Genome Atlas (TCGA) database. Construction of siRNA and overexpression plasmid-packed lentiviral constructs were used to knock down and increase the expression of LINC01117 in LUAD cells. The effect of LINC01117 on LUAD cell migration and invasion was verified by scratch assays and Transwell assays. Western blot assays were performed to verify the effect of knocking down LINC01117 expression on key proteins of the EMT process. The effect of overexpression and knockdown LINC01117 expression on key proteins of the EMT process and the nuclear and cytoplasmic distribution of YAP1, a key effector molecule of the Hippo pathway, was verified by Western blot assays.

### Results

LINC01117 expression was upregulated in LUAD tissues and cell lines. Clinical correlation and prognostic analyses showed that LINC01117 was associated with poorer clinical features (staging and N classification) and poorer prognosis and could be analyzed as an independent prognostic factor. Cell migration and invasion were significantly inhibited in the knockdown group compared to the control group; in contrast, cell migration and invasion

**Competing interests:** The authors have declared that no competing interests exist.

were promoted in the overexpression group. Overexpression of LINC01117 resulted in down-regulation of E-cadherin expression and increased expression levels of N-cadherin, vimentin, ZEB1, snail and slug; in contrast, knockdown of LINC01117 appeared to have the opposite effect. Furthermore, knockdown of LINC01117 increased the enrichment of YAP1 protein in the cytoplasm and reduced its level in the nucleus; overexpression of LINC01117 produced the opposite intracellular distribution results.

## Conclusions

LINC01117 was highly expressed in LUAD, and knockdown of LINC01117 significantly inhibited the migration and invasion of LUAD cells, while overexpression of LINC01117 significantly promoted the migration and invasion of LUAD cells, and affected the EMT process, and was able to alter the distribution of YAP1 in the nucleus and cytoplasm. This suggests that LINC01117 may regulate the activity of the Hippo pathway by altering the nuclear and cytoplasmic distribution of YAP1, which in turn induces the EMT process in lung adenocarcinoma cells and thus exerts a pro-cancer effect. It suggests that LINC01117 may play a key role in the occurrence and development of LUAD.

## Introduction

Lung cancer is a malignant tumors with extremely high morbidity and mortality rates worldwide, seriously affecting human health [1]. According to statistics, there will be approximately 2.2 million new cases and 1.8 million deaths of lung cancer worldwide in 2020, the highest cancer incidence and mortality rate among men and the second highest cancer incidence rate among women, second only to breast cancer [2]. Based on pathology, they can be divided into two main categories: small cell lung cancer and non-small cell carcinoma lung cancer, which can be subdivided into adenocarcinoma of the lung, squamous lung cancer and large cell carcinoma [3]. LUAD is the most prominent tissue subtype of non-small cell lung cancer (NSCLC), with high mortality and metastasis rates [4]. LUAD accounts for 40% of all lung cancers [5] and is commonly found in non-smokers and young women [6], mainly originating from the bronchial mucosal epithelium and bronchial mucus glands, and can occur in the small bronchi or central airways, mostly presenting as peripheral lung cancer, prone to local infiltration and early metastasis [7]. Immunotherapy is often used in advanced stages. Although there have been some improvements in the techniques and tools used to treat LUAD [8, 9], the five-year survival rate for patients is still less than 4% [10, 11].

Most of the RNAs in the human genome cannot be translated into proteins, and these are called non-coding RNAs [12]. Some non-coding RNAs are greater than 200 nucleotides in length and are called long non-coding RNAs (LncRNAs). Among them, long intergenic non-coding RNAs (LncRNAs) are a class of long-stranded non-coding RNAs that lie between coding genes and are usually smaller than protein-coding transcripts [13]. LncRNAs are expressed in low amounts but exhibit strong tissue specificity. Studies have shown that LncRNAs play important regulatory roles in epigenetic, cell cycle and differentiation and other life activities [14, 15].

In this study, a long-stranded intergenic non-coding RNA LINC01117, located on chromosome 2 and 656bp in length, was identified by bioinformatics analysis in the previous phase. Wang et al. found that LINC01117 could be developed as a biomarker for breast cancer

patients through biomarker analysis [16], however, the biological function of LINC01117 in tumors and its molecular mechanism of action have not yet been reported. We found that LINC01117 expression was significantly upregulated in LUAD patient tissues and that high LINC01117 expression was negatively correlated with the prognosis of LUAD patients. We examined the effect of LINC01117 expression on the migration and invasive ability of LUAD cells by knocking down and overexpressing LINC01117, and we investigated the molecular mechanism of LINC01117 in the development of LUAD. In conclusion, LINC01117 as an oncogene promotes the development of lung cancer and through our study LINC01117 is expected to be a biomarker for lung cancer diagnosis and a potential therapeutic target.

## Materials and methods

### Materials

Human non-small cell carcinoma lung cancer cell lines (H1650, H1299, A549, H460) and immortalized normal human bronchial epithelial cells 16HBE are kept in our laboratory. Dulbecco's Phosphate Buffer (DPBS), RPMI-1640, Dulbecco's Modified Eagle Media (DMEM), fetal bovine serum (FBS), 100×Pen Strep, 2.5% trypsin-EDTA were purchased from Gibco, USA. The reverse transcription and qRT-PCR kits were purchased from Takara; Trizol was purchased from ambion; trichloromethane, anhydrous ethanol and isopropanol were purchased from Beijing Chemical Factory; PMSF was purchased from Beijing Solarbio Technology Co. protein loading buffer, 20× protein transfer buffer, 10× SDS-PAGE electrophoresis buffer, and blocking solution were purchased from Beyotime Biotechnology Co. GAPDH, E-cadherin, N-cadherin, and vimentin antibodies were purchased from Cell Signaling Technology (Danvers, MA, USA); RNA-free enzyme water, endotoxin-free plasmid macro extraction kit, and plasmid micro extraction kit were purchased from Beijing TIANGEN Biochemical Co.

### Methods

**Open access data acquisition and analysis.** A dataset including RNA sequencing data from 535 LUAD tumors and 59 non-tumors tissues, as well as other relevant clinical information, was downloaded from The Cancer Genome Atlas website (TCGA, https://gdc-portal.nci.nih.gov/). Differential expression analysis of RNA sequencing data was performed using the Wilcox test with R software. The clinical indicators of the samples in the LUAD cohort, such as gender, age, T, N, M and tumor stage, were classified into two groups. Wilcox test was used to analyze whether the expression level of the above prognostic genes was different between the two groups of different clinical indicators. These were analyzed as independent prognostic factors for LUAD by univariate and multifactorial Cox regression.

**Cell culture.** H1650, H1299 and 16HBE were maintained in 1640 medium and A549 and H460 cells were maintained in DMEM medium, all media were supplemented with 10% fetal bovine serum and penicillin 100 U/mL and streptomycin 100 mg/mL and incubated in a humidified incubator at 37°C and 5% $CO_2$.

**Cell transfection.** LINC01117-siRNA sequence was synthesized by Beijing tsingke Biotechnology. Cells were inoculated into six-well plates the day before transfection and replaced with 2ml of antibiotic-free medium containing 10% FBS on the following day; the transfection system was prepared by adding LINC01117-siRNA to 200ul of transfection buffer to a final concentration of 50mMol/mL, shaking and centrifuging, followed by shaking and centrifuging with 4uL of transfection reagent and incubating for 10min at room temperature. And all of them were added to the cells in the six-well plate, and the control and experimental groups were set up. The transfected cells were incubated in a humidified incubator at 37°C and 5% $CO_2$ for 24h-72h.

**Lentiviral packaging and infection.** The packaging plasmid (psPAX2), envelope plasmid (pMD2.G), and shuttle plasmid containing LINC01117 target gene were synthesized by Beijing tsingke Biotechnology Company. HEK293T cells were inoculated in 100 mm cell culture dishes with $3\times10^6$ cells per dish. 24 h later, when the cells reached 80% confluence, plasmid transfection was performed and plasmids were added according to the instructions (shuttle plasmid: psPAX2: pMD2.G = 4:3:1). After 48 h of transfection, the supernatant was collected and 10 mL of DMEM complete medium was replenished in a Petri dish. The supernatant was centrifuged at 3000 rpm, filtered through a 0.45μm membrane and stored at 4°C. After 24 h, the supernatant was collected again, centrifuged at 3000 rpm and filtered through a 0.45μm membrane. The supernatant was mixed and centrifuged at 100 000 g for 2 h at 4°C. The supernatant was carefully removed and the virus precipitate was suspended using 500μL DMEM and stored frozen at -80°C in a freezer.

**Total RNA extraction and reverse transcription.** The cells were washed with DPBS and the supernatant was aspirated; 500ul of trizol reagent was added to each well, repeatedly blown for 10min at room temperature, Trizol lysate and the lysed cells were transferred together into an ep-tube without RNase, one-fifth of Trizol chloroform was added, i.e. 100uL. Mix well by shaking vigorously for 15s, leave at room temperature for 10 min, centrifuge at 12000×g, 4°C for 15min; remove the upper aqueous phase into a new ep tube, add an equal volume of isopropanol and mix well, leave at room temperature for 10 min, centrifuge at 12000×g, 4°C for 10 min; discard the supernatant, add an equal volume of 75% ethanol as Trizol, wash with gentle blowing and centrifuge at 12000×g, 4°C for 5 min. Centrifuge for 5min; discard supernatant and allow to dry at room temperature; add RNase Free H2O to dissolve the precipitate when it changes from milky white to translucent, mix well and measure the concentration using an ultraviolet photometer. The cDNA was then reverse transcribed and stored at -20°C in the refrigerator.

**Real time PCR.** RNA levels were quantified using SYBR Green PCR Master Mix, with GAPDH as the internal reference gene, and the $2^{-\Delta\Delta CT}$ rule was used to calculate the relative expression of both. qRT-PCR primer sequences were as follows: GAPDH: Forward `GTCTC CTCTGACTTCAACAGCG`, Reverse Sequence `ACCACCCTGTTGCTGTAGCCAA`; LINC01117 Forward: `GAAGUCACUGAGACACCAATT`, LINC01117 Reverse: `UUGGUGUCUCUCAGUGACUUCTT`.

**Wound healing assay.** Cells were inoculated in six-well plates at approximately 4×105 cells per well; the next day the plates were scored perpendicular to the tip of a 10μL gun. The cells were washed 3 times with PBS to remove the scratched cells, medium containing 1% FBS was added and the width of the scratch was recorded under the microscope for 0h and recorded as W0h. The width of the scratch was recorded as W24h after incubation in a humidified incubator at 37°C 5% CO2 for 24h. Mobility was calculated using Image J and plotted using the formula: Mobility = (W0h - W24h)/W0h.

**Transwell assay.** Cells were suspended in serum-free medium and counted, and prepared into 2.5–5×105/ml cell suspension; inoculation of cells: Transwell chambers were placed in 24-well plates, and 200μL of cell suspension was added to the upper chamber (for invasion experiments, matrix gel was spread in the chambers in advance, and the matrix gel was mixed with serum-free medium at a ratio of 1:9 and dropped into the Transwell, the cells were incubated in a 37°C cell incubator for more than 2h, so that the matrix gel was denatured and solidified by heat). 500μL of complete medium containing 10% FBS was added to the lower chamber of the 24-well plate and incubated in a humidified constant temperature incubator at 37°C with 5% CO2 for 24–48 h. The cells were fixed with 4% paraformaldehyde for 15 min and stained with crystal violet for 15 min. The cells and crystalline violet were removed from the upper chamber, dried for a few minutes and observed under the microscope. 3 randomly

selected fields of view were photographed and all the cells in the photographed fields of view were counted, and the number of Transwell in each group of cells was calculated to characterize the migration and invasion ability of the cells.

**Western blot.**  Cytoplasmic/nuclear protein extraction: Cells grown in logarithmic phase were collected by centrifugation, leaving the cell precipitate to be prepared. Add PMSF-added Cytoplasmic Protein Extraction Reagent A. Vortex vigorously for 5s at maximum speed to completely suspend and disperse the cell precipitate. Add Cytoplasmic Protein Extraction Reagent B. Vortex at maximal speed for 5s, ice bath for 1min. Vortex at maximal speed for 5s. Centrifuge and immediately aspirate the supernatant, which is the cytoplasmic protein extracted. For the precipitate, the residual supernatant is completely aspirated and the PMSF-added nucleoprotein extraction reagent is added. The supernatant is immediately aspirated by centrifugation and the nucleoprotein is extracted. Total protein extraction: Cells were lysed on ice with cell lysis solution spiked with protease inhibitor, and cell debris and supernatant were separated by centrifugation at 12,000r/min for 5min at 4˚C. The supernatant was taken and the total protein concentration was determined using the BCA method. The protein samples were added to the appropriate amount of loading buffer, boiled in a water bath for 5min and separated by 7.5%-12% SDS-PAGE, and the membranes were transferred at constant flow for 2h. The PVDF membranes with transferred proteins were closed in fast closing solution for 30min, incubated overnight at 4˚C with proportionally diluted primary antibody, washed three times in TBST and incubated with horseradish peroxidase-associated secondary antibody (1:1000) After washing with TBST, the ECL chemiluminescent solution was reacted with a gel imager, and the results of the protein bands were photographed and the grey scale values were calculated using ImageJ.

**Statistical analysis.**  All data were expressed as mean ± standard deviation (x±s), and the experimental data were statistically processed using the software GraphPad Prism 8.0. Comparisons between multiple groups were made using ANOVA, and comparisons between two groups were processed using t-test. p<0.05 indicates a significant difference with statistical significance.

## Results

### LINC01117 expression was upregulated in LUAD tissues and correlated with aggressive clinical features

According to TCGA data, LINC01117 was significantly upregulated in LUAD tissues (Fig 1A). patients with elevated LINC01117 levels had poorer clinical characteristics, including clinical stage and N classification (Fig 1B and 1C), with higher stage associated with higher LINC01117 expression; patients with elevated LINC01117 levels had significantly shorter overall survival (OS), disease specific survival (DSS) and progression-free interval (PFI) were significantly shorter in patients with elevated LINC01117 levels (Fig 1D–1F, OS, HR = 1.66, p = 0.001; DSS, HR = 1.69, p = 0.006; PFI, HR = 1.39, p = 0.014). Univariate and multivariate analyses indicated that, independent of other clinical characteristics, LINC01117 was a valid prognostic biomarker Table 1.

### LINC01117 is highly expressed in LUAD cells

We examined the role of LINC01117 in LUAD at the cellular level. We used RT-qPCR assays to validate LINC01117 expression levels in non-small cell carcinoma lung cancer and immortalized normal epithelial tissue cells, and showed that LINC01117 expression levels were significantly higher in LUAD cell lines H1650 and A549x than in normal human bronchial

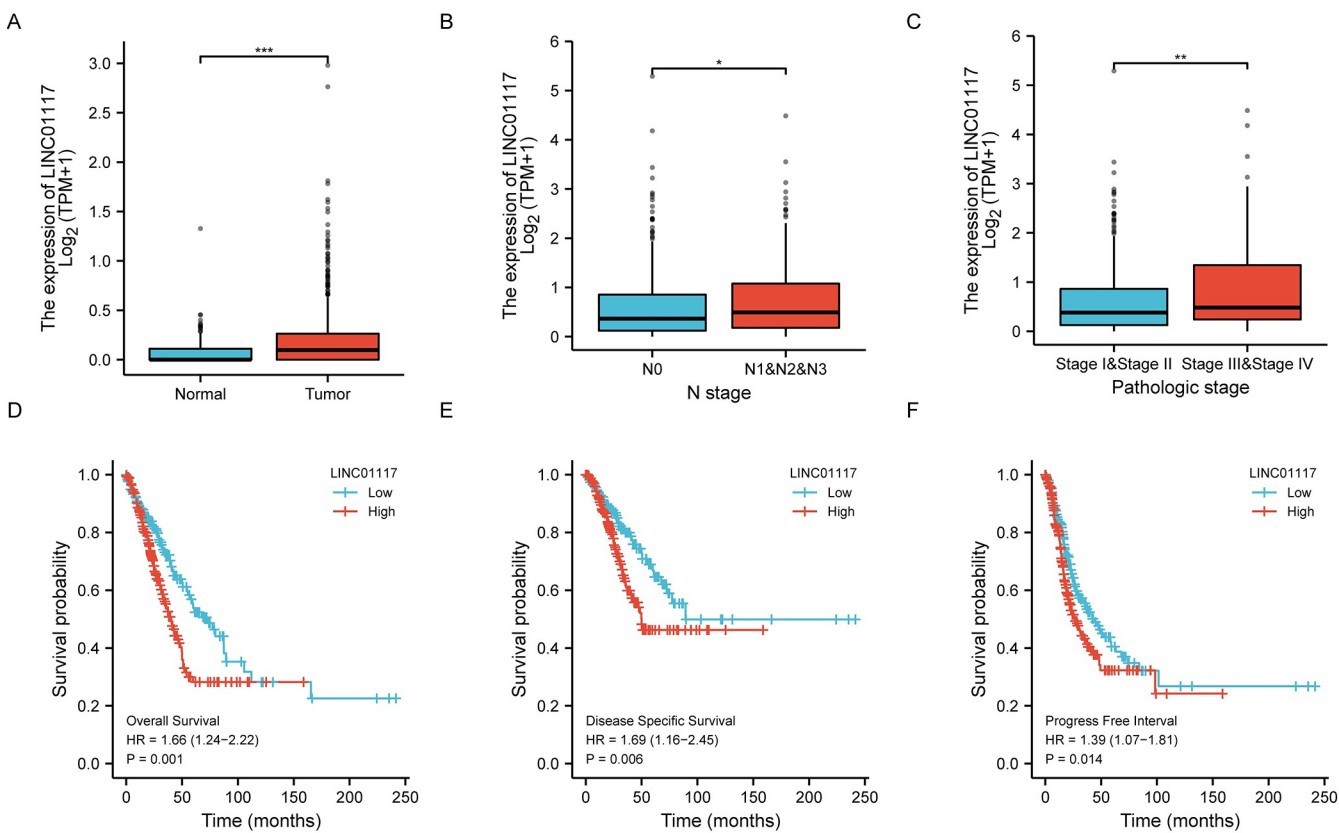

**Fig 1. LINC01117 is upregulated in lung adenocarcinoma tissues and is associated with worse clinical features.** (A) LINC01117 is upregulated in lung adenocarcinoma tissues. (B, C) Association of LINC01117 with clinical staging. (D-F) Association of LINC01117 with patient OS, DSS and PFI.

epithelial cells 16HBE, while in large cell carcinoma H1299 and H460 cell lines were inconsistent in expression (Fig 2A), so we selected LUAD cells H1650 and A549 cells for the follow-up study. The knockdown efficiency of siRNA and overexpression efficiency after lentiviral infection were then verified by RT-qPCR experiments, with knockdown efficiency reaching 70%-80% (Fig 2B and 2D) and overexpression ploidy reaching 3×103 and 3×105 (Fig 2C).

## LINC01117 affects the migration of LUAD cells

Transwell migration assay and wound healing assay were used to examine the changes in cell migration ability after LINC01117 downregulation and overexpression. In the wound healing assay, the cell migration area of the LINC01117 knockdown group was significantly smaller than that of the negative control group (Fig 3A–3D), and the cell migration area of the LINC01117 gene overexpression group was significantly larger than that of the negative control group (Fig 4A–4D); in the Transwell assay, the number of cells migrating to the lower chamber of the LINC01117 knockdown group was significantly smaller than that of the negative In the Transwell assay, the number of cells migrating to the lower chamber of the LINC01117 knockdown group was significantly less than that of the negative control group (Fig 3E and 3F), and the number of cells migrating to the lower chamber of the LINC01117 knockdown group was significantly more than that of the negative control group (Fig 4E and 4F). The results showed that knocking down the expression level of LINC01117 inhibited the in vitro migration of H1650 and A549 cells; on the contrary, the in vitro migration ability of H1650 and A549 cells increased after overexpression of LINC01117.

**Table 1. Univariate and multivariate analyses showed that LINC01117 was an independent prognostic marker for patients with lung cancer.**

| Characteristics | Total(N) | Univariate analysis | | Multivariate analysis | |
|---|---|---|---|---|---|
| | | Hazard ratio (95% CI) | P value | Hazard ratio (95% CI) | P value |
| T stage | 523 | | | | |
| T1&T2 | 457 | Reference | | | |
| T3&T4 | 66 | 2.317 (1.591–3.375) | <**0.001** | 1.804 (1.134–2.869) | **0.013** |
| N stage | 510 | | | | |
| N0 | 343 | Reference | | | |
| N1&N2&N3 | 167 | 2.601 (1.944–3.480) | <**0.001** | 2.244 (1.523–3.307) | <**0.001** |
| M stage | 377 | | | | |
| M0 | 352 | Reference | | | |
| M1 | 25 | 2.136 (1.248–3.653) | **0.006** | 1.547 (0.802–2.987) | 0.193 |
| Pathologic stage | 518 | | | | |
| Stage I&Stage II | 411 | Reference | | | |
| Stage III&Stage IV | 107 | 2.664 (1.960–3.621) | <**0.001** | 1.328 (0.815–2.164) | 0.254 |
| Gender | 526 | | | | |
| Female | 280 | Reference | | | |
| Male | 246 | 1.070 (0.803–1.426) | 0.642 | | |
| Age | 516 | | | | |
| < = 65 | 255 | Reference | | | |
| >65 | 261 | 1.223 (0.916–1.635) | 0.172 | | |
| LINC01117 | 526 | | | | |
| Low | 265 | Reference | | | |
| High | 261 | 1.656 (1.236–2.220) | <**0.001** | 1.828 (1.292–2.585) | <**0.001** |

## LINC01117 affects invasion of LUAD cells

Transwell assays examined changes in cell invasion ability after LINC01117 knockdown and overexpression. The number of H1650 and A549 cells invading into the lower lumen was significantly less in the LINC01117 knockdown group than in the control group (Fig 5A–5D), while the number of H1650 and A549 cells invading into the lower lumen was significantly more in the LINC01117 overexpression group than in the control group (Fig 5E–5H). This indicates that knocking down the expression level of LINC01117 inhibited the invasion of H1650 and A549 cells in vitro; Conversely, the invasion ability of H1650 and A549 cells increased after overexpression of LINC01117.

## LINC01117 affects the EMT process

In tumors metastasis, epithelial mesenchymal transition (EMT) is considered to be one of the important molecular mechanisms. To investigate in depth the molecular mechanisms by which LINC01117 promotes metastasis in LUAD cells, we next explored the role that LINC01117 plays in the EMT process in LUAD. We examined the expression of E-cadherin, N-cadherin, Vimentin, ZEB1, snail and slug, proteins related to the EMT pathway, in A549 cells overexpressing and knocking down LINC01117 by Western blot. The results showed that overexpression of LINC01117 significantly elevated the expression of N-cadherin, Vimentin, ZEB1, snail and slug, while the expression of E-cadherin was inhibited, and on the contrary, the opposite result was obtained in the knockdown group. (Fig 6A and 6B), These experimental results suggest a close correlation between changes in LINC01117 expression and the expression levels of EMT-related markers such as E-cadherin, N-cadherin and Vimentin,

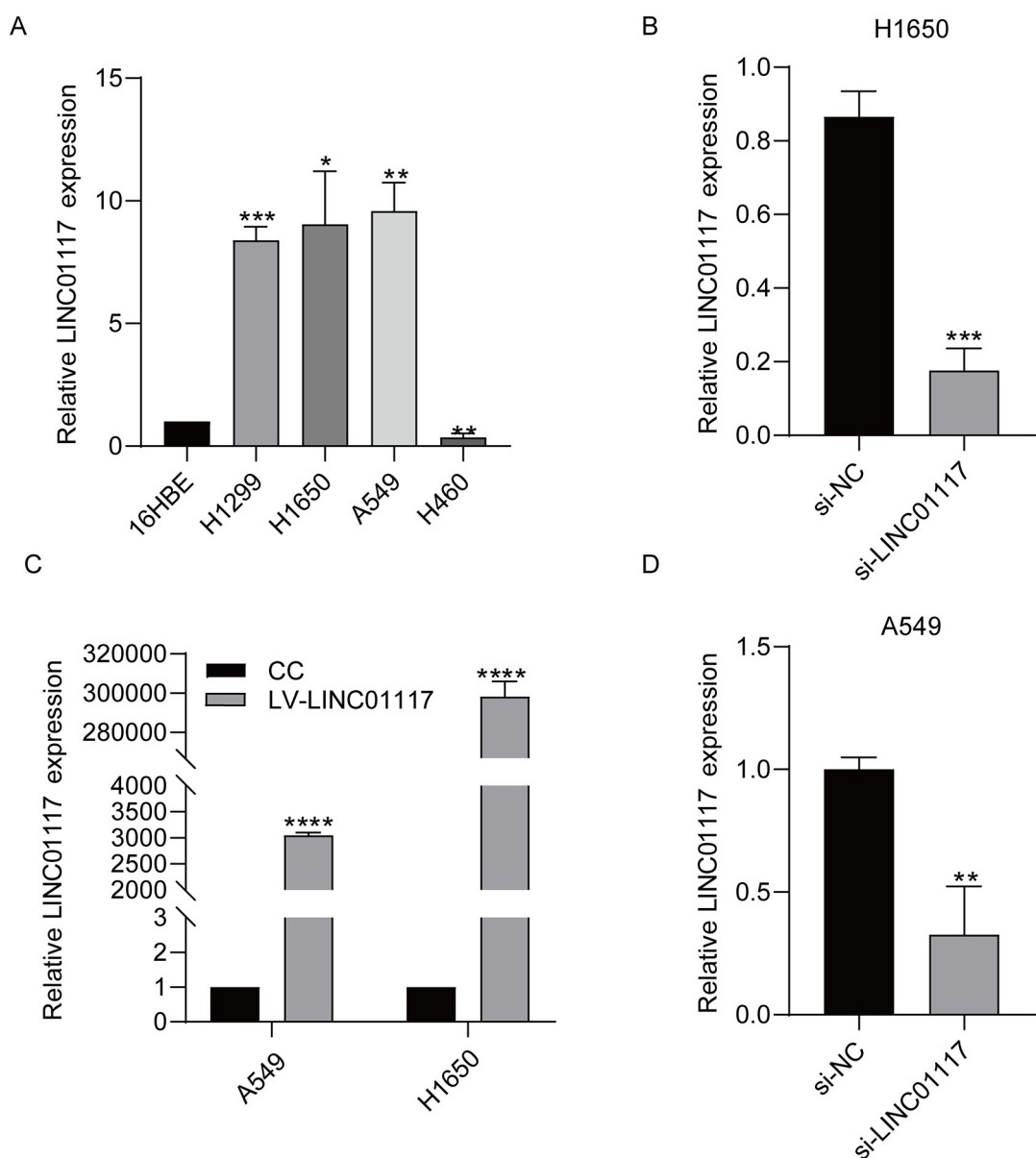

**Fig 2. Expression of LINC01117 in lung adenocarcinoma cells.** (A) Expression of LINC01117 in lung cancer cells. (B) Knockdown efficiency of LINC01117 in 1650 cells. (C) Expression of LINC01117 after overexpression. (D) Knockdown efficiency of LINC01117 in A549 cells.

strongly suggesting that LINC01117 may regulate lung adenocarcinoma progression by affecting EMT process.

**LINC01117 affects the nuclear and cytoplasmic distribution of YAP1 cells.** The Hippo pathway has been widely demonstrated to play an important role in cancer, where YAP1, one of the key components of the Hippo pathway, interacts with EMT-related proteins to influence tumors cell migration and invasion. To investigate the role of LINC01117 in regulating the Hippo pathway in lung adenocarcinoma cells, we extracted total protein, cytoplasmic and nuclear proteins and assayed YAP1 levels in A549 cells with LINC01117 overexpression and knockdown. The results showed that although changes in LINC01117 expression did not

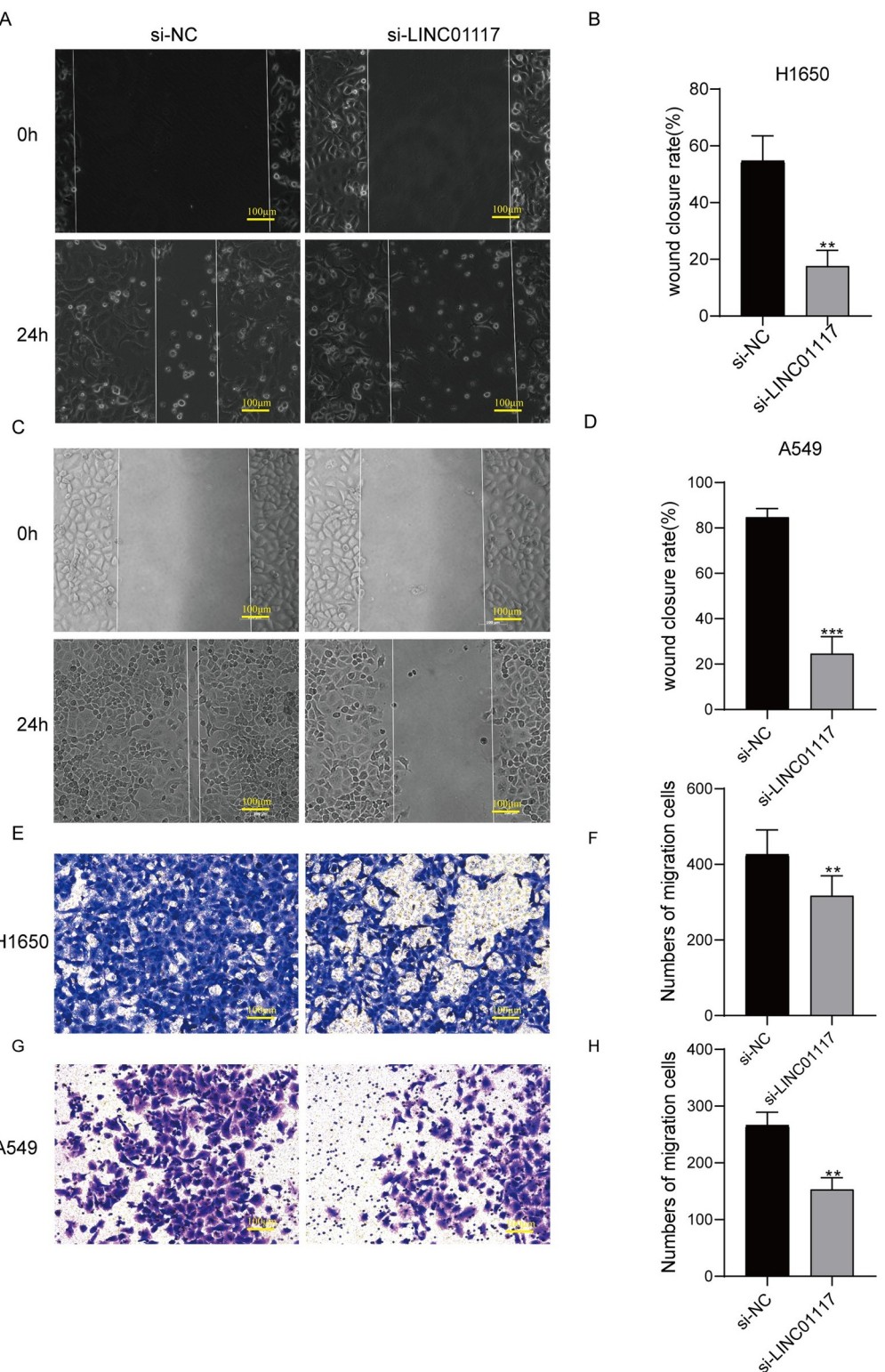

**Fig 3.** LINC01117 knockdown inhibits lung adenocarcinoma cell migration (A, B) Wound healing assay results for H1650 cells. (C, D) Wound healing assay results of A549 cells. (E, F) Transwell assay results of H1650 cells. (G, H) Transwell assay results of A549 cells.

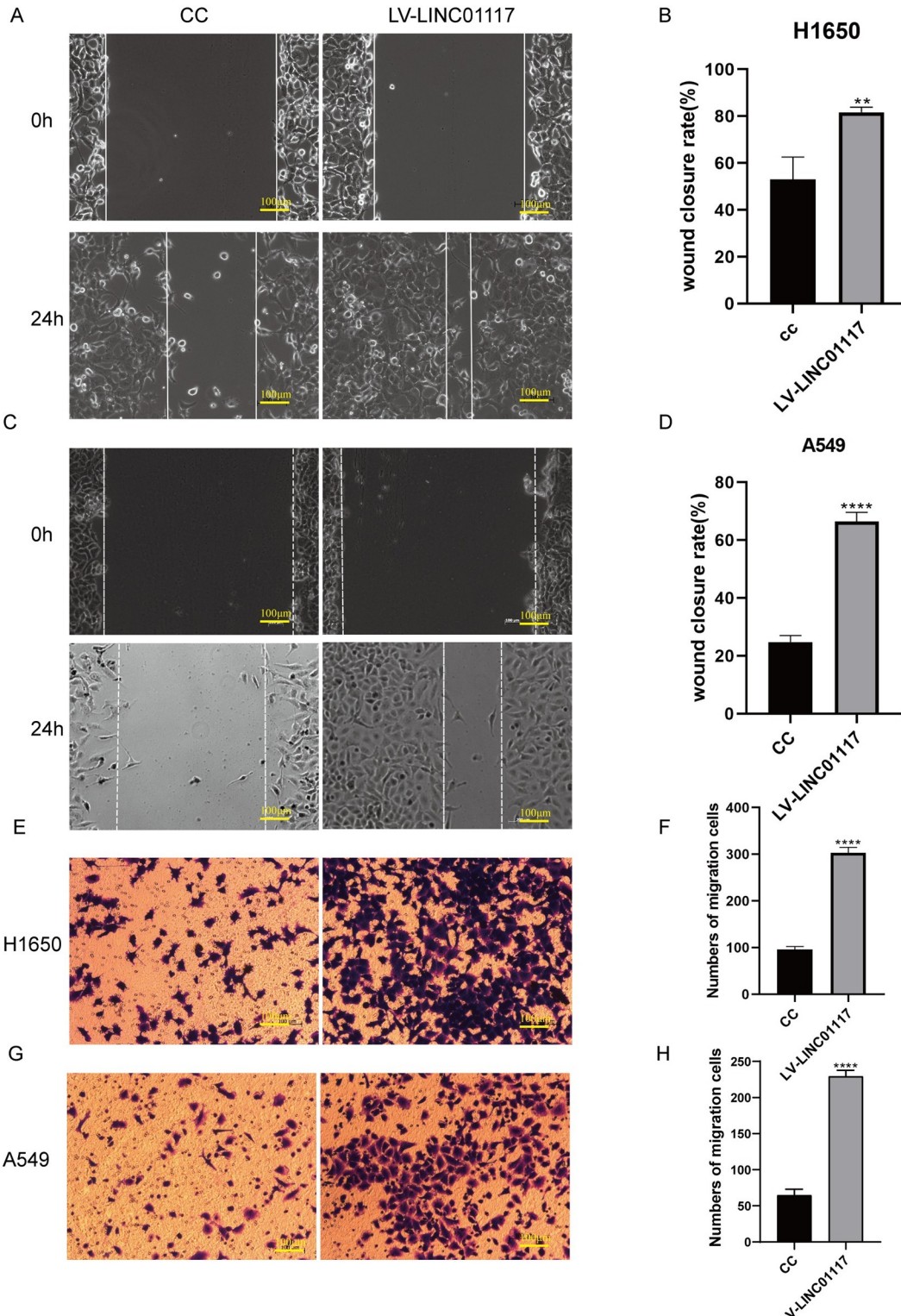

**Fig 4.** LINC01117 overexpression promotes migration of lung adenocarcinoma cells (A, B) Wound healing assay results for H1650 cells. (C, D) Wound healing assay results of A549 cells. (E, F) Results of Transwell assay on H1650 cells. (G, H) Results of Transwell assay on A549 cells.

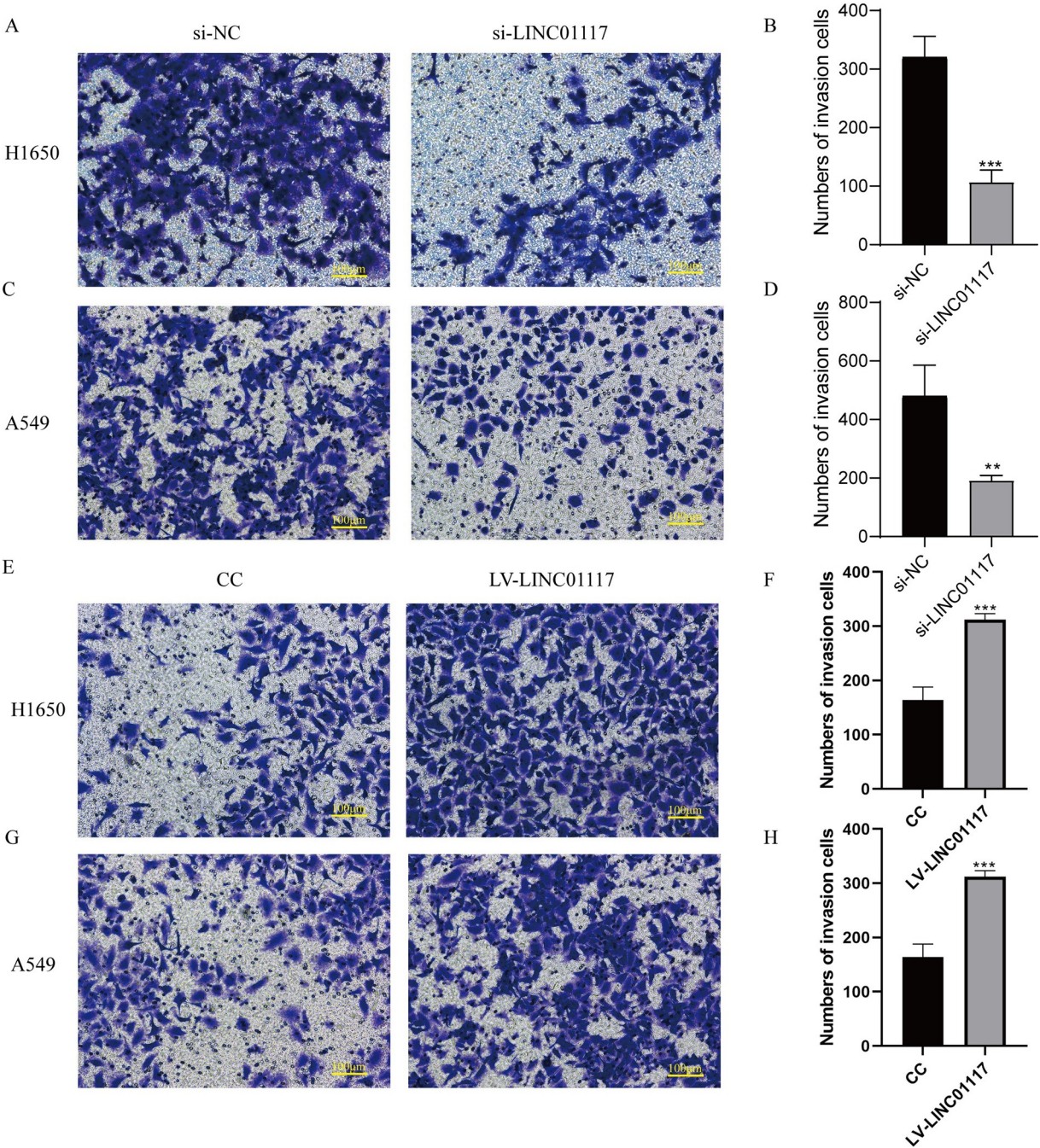

**Fig 5.** LINC01117 promotes lung adenocarcinoma cell invasion (A-D) Transwell invasion of H1650 and A549 cells after knockdown of LINC01117. (E-H) Transwell invasion results of H1650 and A549 cells after overexpression of LINC01117.

significantly affect the overall intracellular levels of YAP1 protein, they significantly altered its distribution in the nucleus and cytoplasm, with knockdown of LINC01117 increasing the enrichment of YAP1 protein in the cytoplasm and decreasing its levels in the nucleus; whereas LINC01117 overexpression produced the opposite intracellular distribution results (Fig 7A and 7B). This suggests that LINC01117 may influence Hippo pathway activity by regulating the distribution of YAP1 in the nucleus and cytoplasm.

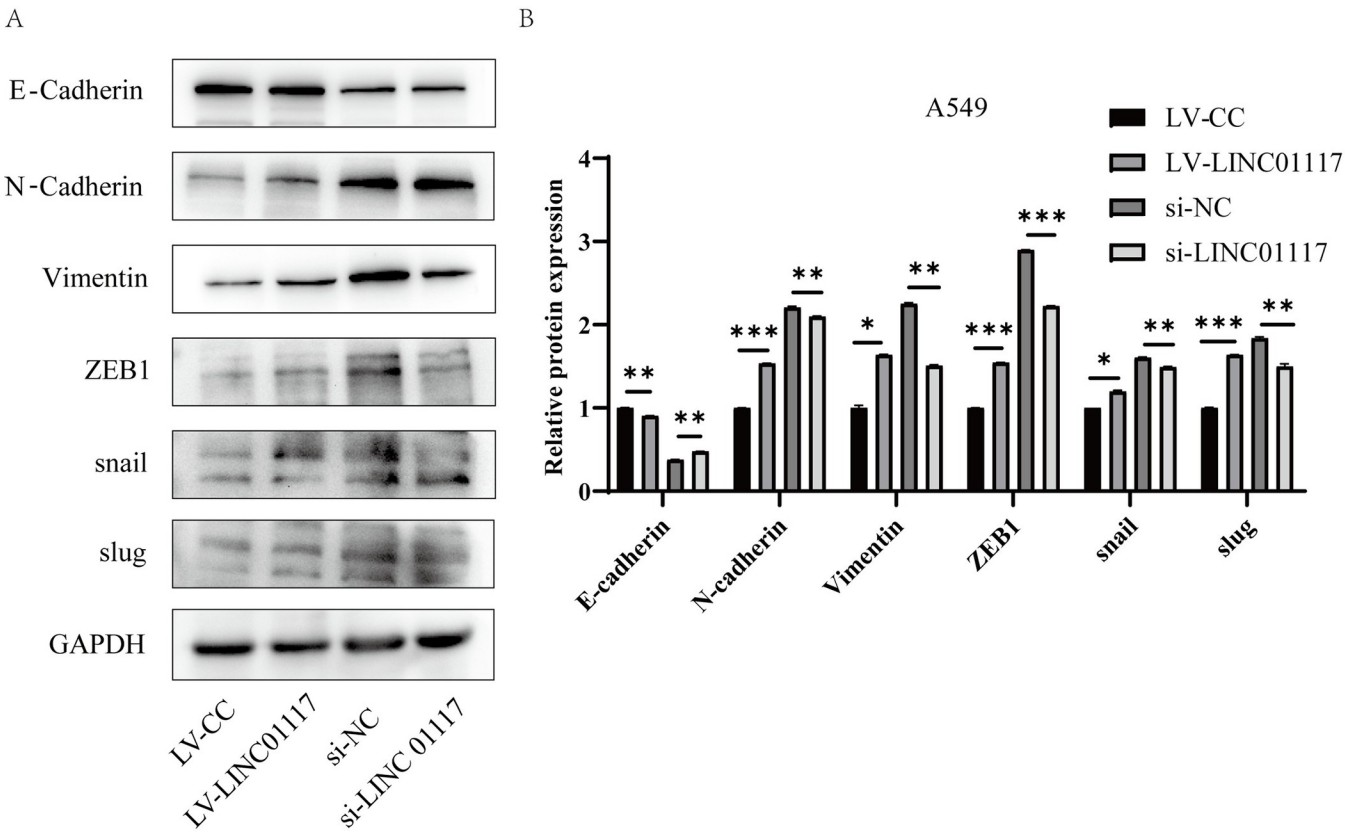

**Fig 6.** LINC01117 affects the EMT process (A, B) Western blot assays examined the expression of key proteins of the EMT process in control and LINC01117 overexpression groups of A549 cells.

## Discussion

LUAD is a highly prevalent malignancy with a trend of increasing morbidity and mortality year by year. Although a combination of surgical treatment, radiotherapy, targeted therapy and immunotherapy has made considerable progress in the treatment of LUAD, metastasis remains the main cause of death in patients with LUAD [17]. Despite the extensive research in the direction of LUAD invasion and metastasis, few stable biomarkers have been identified and used to assess the risk of LUAD metastasis or to predict clinical outcomes. Therefore, the identification of validated key genes associated with LUAD invasion and metastasis is important for early diagnosis and improving the prognosis and clinical outcomes of LUAD patients. In recent years, an increasing number of aberrantly expressed genes have been identified to play a key role in the development and progression of various cancers, and LncRNAs can act as both oncogenes and tumors suppressor genes to regulate tumorigeneses and progression [17, 18]. Therefore, exploring the functions and mechanisms of action of differentially expressed LncRNAs in LUAD may lay the foundation for new diagnostic and therapeutic approaches for LUAD.

EMT is a mechanism that was first discovered in the 1980s [19]. EMT is the transition of cells from an epithelial to a mesenchymal state [20, 21]. This process modifies cell-expressed adhesion molecules, including E-cadherin, which is responsible for tight junctions, and the miRNA200 family, which helps maintain the epithelial phenotype. As cells acquire mesenchymal markers, such as N-cadherin, wave proteins and fibronectin, as well as the fibroblast

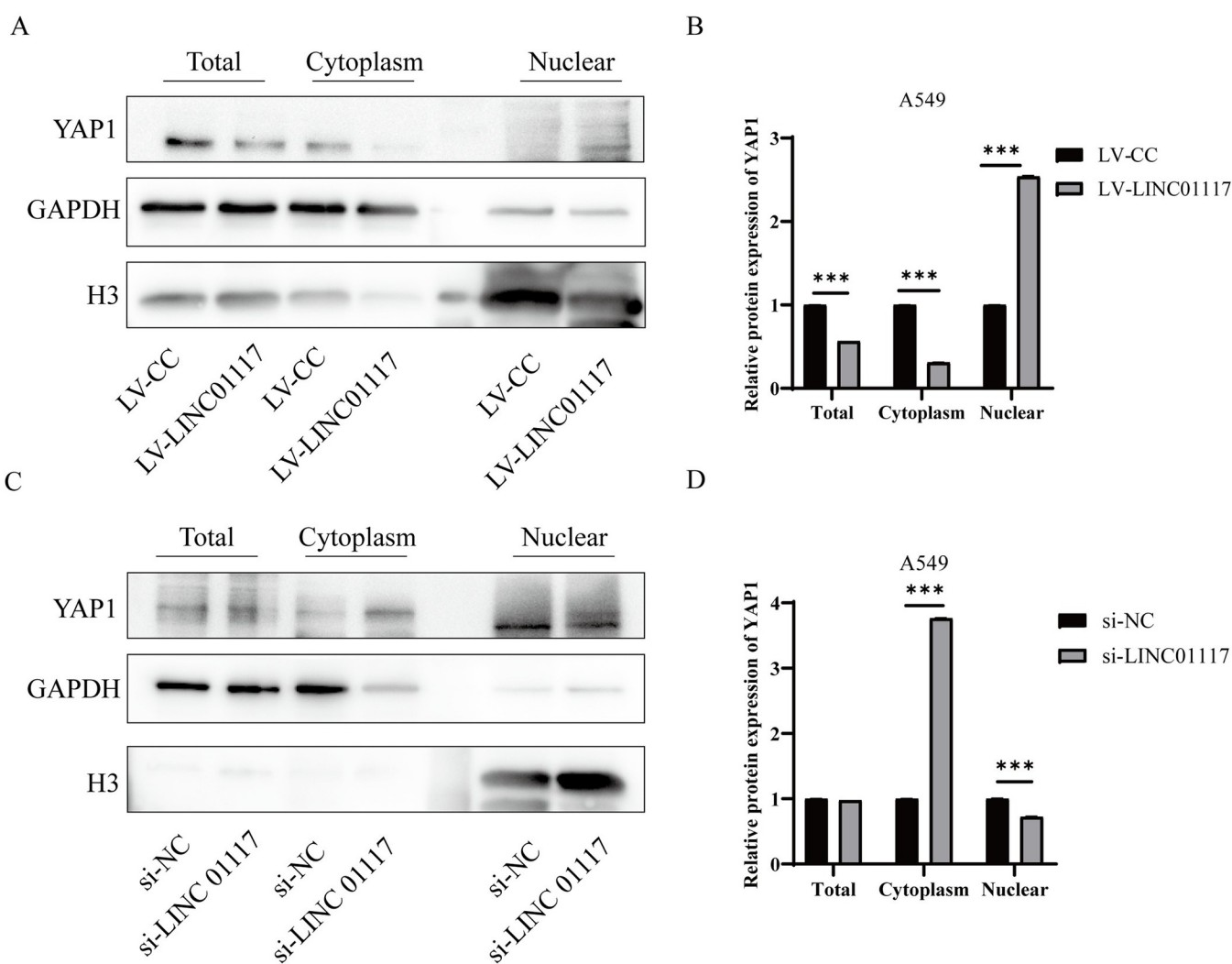

**Fig 7.** Effect of nuclear and cytoplasmic distribution of YAP1 in A549 cells overexpressing and knocking down LINC01117 (A, B) expression of YAP1 in the nucleus and cytoplasm in A549 cells overexpressing LINC01117; (C, D) expression of YAP1 in the nucleus and cytoplasm in A549 cells knocking down LINC01117.

proliferation transcription factors Snail, Slug and Twist, cells migrate towards a more mesenchymal phenotype [22]. EMT can be divided into three distinct types: type 1 EMT is an important physiological event during embryonic development and organogenesis; type 2 EMT is associated with adult tissue regeneration and occurs during wound healing, inducing cell migration, growth and organ fibrosis; and type 3 occurs during cancer progression [23, 24]. The miRNA-200 family and miR-205 together regulate the E-cadherin transcriptional repressors ZEB1 and SIP1, and decreased expression of these miRNAs upregulates ZEB1 and SIP1 [25], inhibits E-cadherin expression, and induces EMT [26]. The EMT process is regulated by multiple signaling pathways and transcriptional mechanisms, including LncRNAs [27]. Hu et al. found more than 99 LncRNAs involved in the EMT process [28]. Wang et al. found that MIR99AHG represses EMT in pulmonary fibrosis via the miR-136-5p/USP4/ACE2 axis [29]. Li et al. validated that LncRNA PCBP1-AS1 inhibited EMT progression suppressing LUAD metastasis [30]. In addition, a number of studies have found that LncRNAs can also regulate the EMT process in tumors through their role in related signaling pathways.

We found corresponding changes in the expression of EMT-related proteins in A549 cells overexpressing and knocking down LINC01117. Overexpression of LINC01117 promoted the EMT process; while knockdown of LINC01117 inhibited the EMT process, suggesting that LINC01117 in lung adenocarcinoma cells may promote cell migration and invasion through the EMT process. The Hippo pathway and the EMT process are inextricably linked [31, 32], and in particular, YAP1, a key effector of the Hippo pathway, can regulate EMT transcription factors through direct or indirect interactions to influence tumors cell invasion and metastasis [33]. In the case of Hippo pathway inhibition, YAP1 enters the nucleus and binds to transcription factors to promote transcriptional expression of downstream genes and influence tumors progression [34]. To investigate the effect of LINC01117 on the Hippo pathway, we extracted total, cytoplasmic and nuclear proteins from A549 cells with overexpression and knockdown of LINC01117, and examined YAP1 expression. The results suggest that LINC01117 may affect the migration and invasion of lung adenocarcinoma cells by influencing the expression of EMT transcription factors through the entry of YAP1 into the nucleus, providing a potential target for the treatment of LUAD.

## Supporting information

**S1 Raw images.**
(PDF)

## Author Contributions

**Conceptualization:** Linjun Liu, Wenjia Ren, Yubai Zhou.

**Data curation:** Linjun Liu, Wenjia Ren.

**Formal analysis:** Licheng Du, Ke Xu, Yubai Zhou.

**Investigation:** Licheng Du, Yubai Zhou.

**Methodology:** Linjun Liu, Wenjia Ren.

**Supervision:** Ke Xu, Yubai Zhou.

**Validation:** Licheng Du.

**Writing – original draft:** Linjun Liu.

**Writing – review & editing:** Ke Xu, Yubai Zhou.

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
