## [Decision Letter · Decision Letter 0]

11 Apr 2023

PONE-D-23-05797LINC01117 inhibits invasion and migration of lung adenocarcinoma through influencing EMT processPLOS ONE

Dear Dr. Liu,

Thank you for submitting your manuscript to PLOS ONE. After careful consideration, we feel that it has merit but does not fully meet PLOS ONE’s publication criteria as it currently stands. Therefore, we invite you to submit a revised version of the manuscript that addresses the points raised during the review process.

We look forward to receiving your revised manuscript.

Kind regards,

Zhiming Li, Ph.D.

Academic Editor

PLOS ONE

Journal Requirements:

In your cover letter, please note whether your blot/gel image data are in Supporting Information or posted at a public data repository, provide the repository URL if relevant, and provide specific details as to which raw blot/gel images, if any, are not available. Email us at plosone@plos.org if you have any questions

Additional Editor Comments (if provided):

Dear Authors,

Your manuscript entitled 'LINC01117 inhibits invasion and migration of lung adenocarcinoma through influencing EMT process' has been evaluated by two referees and their comments are attached below. While the role of LINC01117 in lung cancer is an interesting subject, both reviewers have raised some critical concerns about the manuscript. Specifically, the data quality of figures 4-6 has hampered the interpretation of some of the results. Therefore, we're inviting a major revision so that you can address these concerns before moving forward.

Reviewers' comments:

Reviewer's Responses to Questions

**Comments to the Author**

1. Is the manuscript technically sound, and do the data support the conclusions?

Reviewer #1: No

Reviewer #2: Yes

2. Has the statistical analysis been performed appropriately and rigorously? 

Reviewer #1: No

Reviewer #2: Yes

3. Have the authors made all data underlying the findings in their manuscript fully available?

Reviewer #1: No

Reviewer #2: Yes

4. Is the manuscript presented in an intelligible fashion and written in standard English?

Reviewer #1: No

Reviewer #2: Yes

5. Review Comments to the Author

Reviewer #1: 1. Insufficient innovation in research, and no further research on the downstream regulatory mechanism of linc01117.

2. Is the position of the ruler in Figure 4c inconsistent? Was the image scaled?

3. The protein band in Figure 6 does not match the quantification results. It is recommended to re-quantify using Image J.

4. The band of N-cadherin protein in Figure 6 seems to be a partial cut of a complete band. Please provide the image of original band.

5. The text and reference format does not meet the requirements of this journal.

Reviewer #2: The overall idea of the article is clear, the language is fluent, and the results are clear.Inhibition of LINC01117 expression can significantly inhibit the migration and invasion of lung adenocarcinoma cells. LINC01117 is associated with poor clinical features (stage and N classification) and poor prognosis, suggesting that LINC01117 may play a key role in the occurrence and development of lung adenocarcinoma. The existing deficiencies are as follows:

1. General cell staining was performed for clone formation experiments, and entire Wells of 6-well plates were photographed. In this paper, the entire hole of the 6-well plate is not shown, and no scale is marked on the picture. The experimental images in FIG. 4 are of low quality and are not scaled.

2. As for FIG5, it is too vague to see clearly when enlarged. The cell morphology is not clear and the background is a little messy. It is recommended to optimize the image processing method. In addition, there are too many cells in FIG5A and FIG5E, how to count them accurately? It is recommended to change to a graph with a moderate number of cells.

3. The number of bands in Western blot was too small, so it is suggested to increase the knockdown experiments before trend explanation.

4.The language of this article needs further modification.

Final review: Minor revision

6. PLOS authors have the option to publish the peer review history of their article (what does this mean?). If published, this will include your full peer review and any attached files.

Reviewer #1: No

Reviewer #2: No

---

## [Author Response · Author response to Decision Letter 0]

25 May 2023

Response to Reviewer 1 Comments

Comment 1: Insufficient innovation in research, and no further research on the downstream regulatory mechanism of linc01117. 

Response 1: Thank you for your comment. In our newly submitted article, we complement the downstream regulatory mechanism of LINC01117 in lung adenocarcinoma, which may regulate Hippo pathway activity by altering the distribution of YAP1 in the nucleus and cytoplasm. (Please see lines 329 to 347)

Comment 2: Is the position of the ruler in Figure 4c inconsistent? Was the image scaled? 

Response 2: Thank you very much for your careful identification of the problem with the scale in Figure 4. We have modified the image scales in Figure 4 and scaled the graphics consistently for better layout. We have also reconfirmed the scale of all the images. (Please see Fig 4)

Comment 3: The protein band in Figure 6 does not match the quantification results. It is recommended to re-quantify using Image J.

Response 3: Thank you for your comment. We have performed replicate experiments for Figure 6, in addition to validating the expression of the EMT-related proteins ZEB1, snail and slug proteins, and quantified them using Image J.(Please see Fig 6)

Comment 4: The band of N-cadherin protein in Figure 6 seems to be a partial cut of a complete band. Please provide the image of original band.

Response 4: Thank you for your comment. We have performed a repeat experiment for Figure 6 and uploaded the original uncropped and unadjusted images of all blot or gel result reports as required by the PLOS ONE journal. (Please see Fig 6 and S1_raw_images)

Comment 5: The text and reference format does not meet the requirements of this journal.

Response 4: Thank you for your comment. We deeply apologize for the issue of non-standard formatting in the manuscript. We have reread the manuscript upload requirements for the PLOS ONE journal and made modifications to the parts of the text and reference formats that do not comply with the requirements of this journal. (See highlighted sections in the text) 

Response to Reviewer 2 Comments

The overall idea of the article is clear, the language is fluent, and the results are clear. Inhibition of LINC01117 expression can significantly inhibit the migration and invasion of lung adenocarcinoma cells. LINC01117 is associated with poor clinical features (stage and N classification) and poor prognosis, suggesting that LINC01117 may play a key role in the occurrence and development of lung adenocarcinoma. The existing deficiencies are as follows:

Comment 1: General cell staining was performed for clone formation experiments, and entire Wells of 6-well plates were photographed. In this paper, the entire hole of the 6-well plate is not shown, and no scale is marked on the picture. The experimental images in FIG. 4 are of low quality and are not scaled.

Response 1: Thank you for your comment. The clone formation experiment is not involved in this manuscript, and the staining images you see may be the experimental results of cell invasion and migration. We apologize for our carelessness regarding the lack of scale in the images. We have re standardized the scale and size of the images and adjusted their clarity. (Please see Fig 4)

Comment 2: As for FIG5, it is too vague to see clearly when enlarged. The cell morphology is not clear and the background is a little messy. It is recommended to optimize the image processing method. In addition, there are too many cells in FIG5A and FIG5E, how to count them accurately? It is recommended to change to a graph with a moderate number of cells.

Response 2: Thank you for your comment. Based on your suggestion, we have redone the experiment in Figure 5 to address issues such as unclear images and excessive cell density, and adjusted the clarity of the experimental results. As you mentioned, in the new experimental results graph, a moderate cell density is indeed more conducive to accurate counting. (Please see Fig 5)

Comment 3: The number of bands in Western blot was too small, so it is suggested to increase the knockdown experiments before trend explanation.

Response 3: Thank you for your comment. Based on your suggestion, we conducted knockdown experiments on LINC01117 and validated the levels of key proteins related to the EMT process through Western blot experiments. The results met our expectations and demonstrated that LINC01117 activated the EMT pathway. In addition, we validated the protein expression of other EMT related molecules ZEB1, Snail and Slug, and supplemented the above experimental results in the manuscript, increasing the credibility of our conclusion. (Please see Fig 6)

Comment 4: The language of this article needs further modification.

Response 4: Thank you for your comment. We deeply apologize for the language used in the manuscript. We spent a long time revising the manuscript and have highlighted it in yellow in the text. We have now conducted research on language and readability, and we really hope to have a substantial improvement in the language proficiency of the article.

---

## [Decision Letter · Decision Letter 1]

15 Jun 2023

LINC01117 inhibits invasion and migration of lung adenocarcinoma through influencing EMT process

PONE-D-23-05797R1

Dear Dr. Liu,

We’re pleased to inform you that your manuscript has been judged scientifically suitable for publication and will be formally accepted for publication once it meets all outstanding technical requirements.

Kind regards,

Zhiming Li, Ph.D.

Academic Editor

PLOS ONE

Additional Editor Comments (optional):

Reviewers' comments:

Reviewer's Responses to Questions

**Comments to the Author**

1. If the authors have adequately addressed your comments raised in a previous round of review and you feel that this manuscript is now acceptable for publication, you may indicate that here to bypass the “Comments to the Author” section, enter your conflict of interest statement in the “Confidential to Editor” section, and submit your "Accept" recommendation.

Reviewer #2: All comments have been addressed

2. Is the manuscript technically sound, and do the data support the conclusions?

Reviewer #2: Yes

3. Has the statistical analysis been performed appropriately and rigorously? 

Reviewer #2: Yes

4. Have the authors made all data underlying the findings in their manuscript fully available?

Reviewer #2: Yes

5. Is the manuscript presented in an intelligible fashion and written in standard English?

Reviewer #2: Yes

6. Review Comments to the Author

Reviewer #2: In this study, authors LINC01117 expression was upregulated in LUAD tissues and

correlated with aggressive clinical features. This study is relatively clear, the English language is standardized, the research purpose is clear, and the problems found in the past have been corrected accordingly.

7. PLOS authors have the option to publish the peer review history of their article (what does this mean?). If published, this will include your full peer review and any attached files.

Reviewer #2: **Yes: **Chuanxin Wang

---

## [Editor Report · Acceptance letter]

21 Jun 2023

PONE-D-23-05797R1 

LINC01117 inhibits invasion and migration of lung adenocarcinoma through influencing EMT process 

Dear Dr. Liu:

I'm pleased to inform you that your manuscript has been deemed suitable for publication in PLOS ONE. Congratulations! Your manuscript is now with our production department. 

Kind regards, 

on behalf of

Dr. Zhiming Li 

Academic Editor

PLOS ONE